# Self-Assembling E2-Based Nanoparticles Improve Vaccine Thermostability and Protective Immunity against CSFV

**DOI:** 10.3390/ijms25010596

**Published:** 2024-01-02

**Authors:** Hetao Song, Sahibzada Waheed Abdullah, Chenchen Pei, Xiaoni Shi, Xiangyang Chen, Yuqing Ma, Shuanghui Yin, Shiqi Sun, Yong Huang, Huichen Guo

**Affiliations:** 1College of Veterinary Medicine, Northwest A&F University, Yangling 712100, China; sht854844223@sina.com; 2State Key Laboratory for Animal Disease Control and Prevention, College of Veterinary Medicine, Lanzhou University, Lanzhou Veterinary Research Institute, Chinese Academy of Agricultural Sciences, Lanzhou 730046, China; waheed_149@yahoo.com (S.W.A.); shixiaoni001@126.com (X.S.); m15966290872_1@163.com (X.C.); m838152546@163.com (Y.M.); yinshuanghui@caas.cn (S.Y.); sunshiqi@caas.cn (S.S.); 3Gansu Province Research Center for Basic Disciplines of Pathogen Biology, Lanzhou 730046, China

**Keywords:** nanoparticles, vaccine, E2, thermostability, protective immunity, CSFV

## Abstract

Classical swine fever virus (CSFV) is a highly contagious pathogen causing significant economic losses in the swine industry. Conventional inactivated or attenuated live vaccines for classical swine fever (CSF) are effective but face biosafety concerns and cannot distinguish vaccinated animals from those infected with the field virus, complicating CSF eradication efforts. It is noteworthy that nanoparticle (NP)-based vaccines resemble natural viruses in size and antigen structure, and offer an alternative tool to circumvent these limitations. In this study, we developed an innovative vaccine delivery scaffold utilizing self-assembled mi3 NPs, which form stable structures carrying the CSFV E2 glycoprotein. The expressed yeast E2-fused protein (E2-mi3 NPs) exhibited robust thermostability (25 to 70 °C) and long-term storage stability at room temperature (25 °C). Interestingly, E2-mi3 NPs made with this technology elicited enhanced antigen uptake by RAW264.7 cells. In a rabbit model, the E2-mi3 NP vaccine against CSFV markedly increased CSFV-specific neutralizing antibody titers. Importantly, it conferred complete protection in rabbits challenged with the C-strain of CSFV. Furthermore, we also found that the E2-mi3 NP vaccines triggered stronger cellular (T-lymphocyte proliferation, CD8^+^ T-lymphocytes, IFN-γ, IL-2, and IL-12p70) and humoral (CSFV-specific neutralizing antibodies, CD4^+^ T-lymphocytes, and IL-4) immune responses in pigs than the E2 vaccines. To sum up, these structure-based, self-assembled mi3 NPs provide valuable insights for novel antiviral strategies against the constantly infectious agents.

## 1. Introduction

Classical swine fever (CSF) is a highly contagious and fatal disease of pigs that is caused by classical swine fever virus (CSFV) and is considered one of the most devastating diseases for the global pig industry [1,2]. CSFV belongs to the genus pestivirus within the *flaviviridae* family [1]. The CSFV genome is a single-stranded, positive-sense RNA virus of about 12.3 kb, which contains a single open reading frame (ORF) and encodes a polyprotein that processes four structural proteins (C, E^rns^, E1, and E2) and eight non-structural proteins (N^pro^, p7, NS2, NS3, NS4A, NS4B, NS5A, and NS5B) [3]. The E2 glycoprotein is the most immunogenic among all of the CSFV proteins and induces the production of high levels of neutralizing antibodies, providing protection against lethal CSFV challenge [4,5]. E2 is the preferred target of CSF subunit vaccine research [6] and has been expressed in baculovirus [7] and yeast [8] expression systems.

Currently, immunization with vaccines is a major strategy for preventing and controlling CSF. Although conventional inactivated or attenuated live vaccines are efficacious in the control or elimination of CSF, they present limitations, one of which is that they do not allow for serological differentiation between infected and vaccinated animals (DIVA), posing a severe challenge to the control and eradication of CSF [9,10]. Therefore, alternative vaccination strategies that allow DIVA are urgently necessary.

Virus-like particles (VLPs), a type of self-assembled nanoparticle (NP), have achieved significant breakthroughs in vaccine development that could compensate for the disadvantages of traditional inactivated or attenuated live vaccines [11,12]. VLPs are empty particles that are similar to viruses and are formed by assembling one or more recombinant expressed viral structural proteins [13]. VLPs share high similarity with the viral structure, but they do not contain viral genetic material and lack the capacities for replication and infectivity. However, they can activate the host’s natural and acquired immune responses [13,14,15]. Importantly, compared with an inactivated vaccine or single viral protein or polypeptide, VLPs have advantages in terms of cost, safety, efficiency, and disease purification [16,17]. It should be noted that the potential of VLPs in vaccine development has been well-demonstrated in the commercialization of human papillomavirus, hepatitis B virus, and porcine circovirus vaccines [16,18]. So far, VLPs are considered to be the vaccine candidate with the most potential. They are poised to replace traditional inactivated vaccines due to their superior immunogenicity and capacity to distinguish infected animals from immunized animals to eradicate CSF [16]. Despite these successes, challenges remain, particularly in the formation of VLPs from certain viral antigens such as the CSFV E2 protein, which limits vaccine efficacy. Thus, ameliorations are required in order to raise the protective immunity of vaccines, and NP-based delivery technology may offer a powerful alternative tool. Usually, target antigens were linked to NPs through chemical modification or gene fusion. With NPs as delivery scaffolds, multiple copies of the target antigens can be displayed on the surface of NPs with correct conformation [12,19]. Previous studies have shown that i301 and mi3, based on the computational design of an icosahedral nanocage, were capable of self-arranging. They spontaneously form a highly ordered 60-subunit dodecahedral NP, which serves as a promising alternative for novel NP-based vaccine design [20,21]. Liu et al. reported the development of self-assembled mi3 NP DIVA vaccines carrying the CSFV E2 glycoprotein using a Bac-to-Bac system, which demonstrated effective protection in the lethal challenge test of CSFV [11,12]. In addition, the scaffold protein of NPs could be a useful DIVA marker by detecting the antibody against scaffold protein. These self-assembled NPs have become the focus of modern vaccine research because of their antigen stability, are highly immunogenic, have good histocompatibility, and easy modification [11,12,19,22,23].

To sum up, developing a safe and effective vaccine allowing DIVA is very important. Our study focused on enhancing the protective efficacy of vaccines through NP-based technology. Hence, self-assembled NPs were developed and characterized as a vaccine delivery scaffold. We constructed recombinant plasmids with CSFV E2 fused to mi3 protein expressed in the PichiaPink system. Our investigations into the self-assembly efficiency and storage stability of E2-mi3 NPs suggest promising potential for vaccine development. Notably, E2-mi3 NP vaccines conferred complete protection in rabbits against CSFV C-strain challenges and triggered robust protective immunity in pigs. These results encourage further work toward the development of self-assembled mi3 NP vaccines against CSFV.

## 2. Results

### 2.1. Expression and Characterization of CSFV E2-mi3 NPs

To investigate the self-assembly and antigenicity of the CSFV E2-mi3 protein, E2, mi3, and E2-mi3 expression plasmids were generated as shown in Figure 1A,C,E. The fusion proteins were successfully overexpressed in the PichiaPink system and purified using nickel ion affinity chromatography at room temperature. All of the proteins bands were consistent with the expected sizes, as explained by SDS-PAGE and Western blot analysis (Figure 1B,D,F), suggesting that the E2, mi3, and E2-mi3 proteins were successfully expressed. Next, NP assembly was carried out and examined by DLS and TEM. The results revealed that fusion proteins mi3 and E2-mi3 could self-assemble into uniform NPs, which had an approximate diameter of 25–30 nm (Figure 1G–J). Collectively, these results demonstrate the effective production of E2, mi3 NPs, and E2-mi3 NPs using the PichiaPink expression system.

### 2.2. Analysis of CSFV E2-mi3 Antigenic Epitopes 

To identify whether the mi3 skeleton protein has an impact on the antigenic epitopes of CSFV E2 protein, the locations of the identified conformational epitopes in the CSFV E2 protein and E2-mi3 protein were analyzed using the Pymol (2.5) online analysis software. The prediction of tertiary structures revealed that the major regions of the E2 and E2-mi3 proteins, excluding the mi3 skeleton protein, exhibited substantial overlap. Notably, the interaction site between the E2 protein and the mi3 skeleton protein did not coincide with the antigenic epitopes of the E2 protein, as depicted in Figure 2. These results indicate that the E2 protein has a certain affinity with the mi3 skeleton protein, and that the mi3 skeleton protein has no effect on the antigenic epitopes of the E2 protein. The citations for the antigenic epitopes of CSFV E2 protein are listed in [24,25,26].

### 2.3. E2-mi3 NPs Exhibited Robust Thermostability and Long-Term Storage Stability

To test the thermostability of the NPs, purified E2-mi3 NPs were incubated in NP assembly solution at temperatures ranging from 25 to 70 °C for 1 h. Aggregates were removed by centrifugation, and soluble fractions in supernatant were quantified via densitometry. It was observed that at temperatures up to 70 °C, a minimum of 51% of the E2-mi3 NP protein remained soluble. Exposure to 70 °C resulted in the aggregation and subsequent loss of 83.2% of the E2 protein (Figure 3A). At elevated temperatures, a small increase was observed in the size distribution of the E2 mi3-NPs protein, as measured by DLS (Figure 3B).

Furthermore, the study extended to assess the room temperature (25 °C) stability of the nanoparticles. Over intervals of 2, 4, and 6 weeks, the integrity of the E2-mi3 NPs was scrutinized using TEM. The findings showed that the percentage of intact E2-mi3 NPs was over 80% for 4 weeks and up to 50% at 6 weeks (Figure 3C). In addition, the effective antigens were quantitatively measured by Dot blotting. We found that the degradation rates of the E2-mi3 NP protein were significantly slower than the E2 protein (Figure 3D). To sum up, the E2-mi3 NP protein exhibited robust thermostability and long-term storage stability.

### 2.4. E2-mi3 NPs Promoted Cellular Uptake by RAW264.7 Cells

Higher antigen uptake efficiency leads to better antigen specific immune responses. To determine whether E2-mi3 NPs also have the ability to promote uptake into cells, RAW264.7 cells were incubated with the E2 protein and E2-mi3 NP protein for different periods of time (1, 2, 4, 6, and 8 h). Subsequently, the cellular uptake of both the E2 protein and E2-mi3 NP protein was assessed through Western blot analysis. As shown in Figure 4, the amount of E2 protein (peaked at 6 h) and E2-mi3 NP protein (peaked at 4 h) taken up by the cells gradually increased with incubation time. Interestingly, the uptake efficiency of the E2-mi3 NP group was markedly higher (*p* < 0.05) than that of the E2 group, which demonstrated that the formation of NPs promotes the entry of antigen into cells.

### 2.5. Enhanced Specific Antibody Responses Induced by E2-mi3 NPs

For rabbit immunization: The E2-mi3 NP group and E2 group were intramuscularly inoculated with 10 μg E2-mi3 NPs or 10 μg E2, respectively. The same amount of phosphate-buffered saline (PBS) was given to the rabbits in the PBS group by the same method. For swine immunization: The E2-mi3 NP group and E2 group were injected intramuscularly with 40 μg E2-mi3 NPs or 40 μg E2, respectively. Pigs in the PBS group were injected with PBS. Serum samples were collected from the rabbits and pigs at 7, 14, 21, and 28 days post-immunization (dpi). 

The efficacy of E2-mi3 NPs to induce an in vivo protective response was further evaluated in rabbits and pigs. Specific antibodies against CSFV E2 were measured by blocking ELISA (CSFV antibody positive: sample blocking ratio% ≥ 40%). As shown in Figure 5A,B, compared with the PBS group, the specific antibody levels in the E2 group and E2-mi3 NP group were dramatically increased (*p* < 0.05) at 14, 21, and 28 dpi. It was noticed that the E2-mi3 NP group significantly elevated (*p* < 0.05) specific antibody levels than those of the E2 group at 14, 21, and 28 dpi. Subsequently, the levels of mi3 NP specific antibodies were detected through indirect-ELISA. Of note, a higher level of specific antibodies against mi3 NPs was observed in the E2-mi3 NP group (*p* < 0.05), other than the E2 group and PBS group at 14, 21, and 28 dpi (Figure 5C,D). These results explain that in the E2-mi3 NP group, antibodies against mi3 NPs did not interfere with the antibodies targeting E2.

### 2.6. Neutralizing Ability of E2-mi3 NPs Immunized Serum

The rabbit and pig sera collected at 28 dpi were diluted with physiological saline at a ratio of 1:4, 1:16, and 1:64. The diluted sera were mixed with an equal volume of the CSFV C-strain. In the E2 group and E2-mi3 NP group, each rabbit was inoculated with 1 mL of serum–virus mixtures through the ear vein. Meanwhile, each rabbit in the PBS group received 1 mL of physiological saline–virus mixtures using the same method. The results, as presented in Table 1, revealed that the rabbit or pig serum from the PBS group did not exhibit virus-neutralizing activities at the lowest dilution tested (1:4). In contrast, the rabbit or pig serum from the E2 group and E2-mi3 NP group potently neutralized the virus. However, only three out of five rabbits in the E2 group exhibited neutralization at the 1:16 dilution. On the other hand, all rabbits in the E2-mi3 NP group showed strong neutralization at the same dilution. Importantly, four out of five rabbits in the E2 mi3 NP group successfully neutralized the C-strain viruses at the 1:64 dilution, while no neutralization was observed in the E2 group. Based on these results, it can be concluded that E2-mi3 NP vaccines against CSFV dramatically improved the CSFV-neutralizing antibodies.

### 2.7. E2-mi3 NPs Induced Complete Protection from CSFV C-Strain Challenge

To evaluate the immunogenicity of E2-mi3 NP vaccines, an immunization and challenge experiment were conducted in the rabbits. As shown in Figure 6 and Table 2, from 12 h after CSFV C-strain challenge, the PBS group exhibited acute fever (40~41.7 °C) and other clinical symptoms including inappetence, depression, and incoordination (protection rate: 0%). Two out of five rabbits in the E2 group presented a short-term fever (40~41.5 °C), but all returned to normal 48 h later (protection rate: 60%). Importantly, following C-strain challenge, no febrile response and other clinical signs were observed in the E2-mi3 NP group (protection rate: 100%). Overall, the above challenge results indicate that the E2-mi3 NP vaccines were able to confer complete protection of rabbits from challenge with the C-strain.

### 2.8. Virological Protection of Vaccinated Rabbits from CSFV C-Strain Challenge

The anticoagulated blood samples were collected at 0, 1, 2, 3, and 4 days post-challenge and CSFV RNA was quantified by RT-qPCR. The results are shown in Figure 7. At 1 day post-challenge, all rabbits in the PBS group were detected positive for CSFV RNA, with the RNA loads approximately 10^5^ copies/μL, and the RNA level peaked at 2 days post-challenge. In contrast, viral RNA was tested in two out of five rabbits in the E2 group at 1 day post-challenge, with the RNA loads over 10^3^ copies/μL, up to a peak load of over 10^5^ copies/μL. It was observed that viral RNA was not detected in the E2-mi3 NP group at different time points post-challenge. These results further validate that the E2-mi3 NP vaccines effectively suppressed or neutralized virus in the blood.

### 2.9. E2-mi3 NPs Reduced Viral Replication in Rabbits following CSFV C-Strain Challenge

The tissues (hearts, livers, spleens, lungs, and kidneys) were collected from rabbits at 4 days post-challenge. CSFV antigens in the tissues were examined by immunohistochemistry (IHC). In Figure 8A, the IHC analysis showed that CSFV antigens (brown granules) were diffusely distributed in these tissues of the PBS group. Additionally, slightly fewer CSFV antigens were detected in the E2 group. In contrast, no obvious viral antigens were observed in these tissues of the E2-mi3 NP group. Then, the proportions of the IHC-positive cell area were calculated with the HALO digital pathology system. As shown in Figure 8B, the proportions of positive cell area in the E2 group and E2-mi3 NP group were significantly lower (*p* < 0.05) than those in the PBS group. It should be noted that the values in the E2-mi3 NP group were dramatically decreased (*p* < 0.05) in comparison to the E2 group. Taken together, these results indicate that the E2-mi3 NP vaccines suppressed the production of CSFV in vivo, rendering the rabbits less susceptible to CSFV infection.

### 2.10. Observation of Pathological Injuries following CSFV C-Strain Challenge

The tissues (hearts, livers, spleens, lungs, and kidneys) were collected from rabbits at 4 days post-challenge. At 1~2 days post-challenge, the body weight in the PBS group and the E2 group showed a decreasing tendency compared with the E2-mi3 NP group. At 3 days post-challenge, the PBS group and the E2 group began to regain body weight. It was noticed that the body weight of rabbits in the E2-mi3 NP group showed no significant difference during the whole challenge experiment (Appendix A). Macroscopically, the PBS group showed mild clinical lesions in different tissues such as a larger size than the E2 group and E2-mi3 NP group. Furthermore, the spleens of the PBS group were dark or the texture became hard. In contrast, the E2 group and E2-mi3 NP group exhibited a normal macroscopic structure, except that the E2 group exhibited a larger size than the E2-mi3 NP group (Appendix A); these results were consistent with the organ weight and organ index shown in Appendix A. After C-strain challenge, increased (*p* < 0.05) organ weight and organ index were detected in the PBS group compared with the E2 group and E2-mi3 NP group, except for the heart and liver (Appendix A).

After C-strain challenge, the PBS group infiltrated large numbers of inflammatory cells, which led to a significant increase in organ weight and organ index. Corresponding to the above challenge results, the features were also supported by histopathological analysis. Microscopically, as depicted in Appendix A, several histopathological changes were observed in the tissues of rabbits belonging to the PBS group. Specifically, in the hearts, signs of myocardial fiber degeneration and necrosis were evident. Additionally, the livers displayed indications of lymphocyte infiltration, hepatocyte degeneration, and necrosis. In the spleens, observable alterations included splenic sinusoid dilatation, red pulp congestion, and erythrocyte aggregation. Moreover, the lungs exhibited thickening of the alveolar walls accompanied by a reduction in airspace areas. Finally, notable dilatation of renal tubules, epithelial cell degeneration, and interstitial fibrous tissue hyperplasia were observed. Furthermore, histopathological changes in the E2 group were similar to that in the PBS group, except that the degree of pathological injuries was lesser. It is noteworthy that there were no histopathological changes observed within the E2-mi3 NP group. As indicated in Appendix A, the histologic scoring showed the difference in the severity of pathological injuries among the three groups after C-strain challenge. These results suggest that E2-mi3 NP vaccines conferred complete protection to rabbit organ tissues against CSFV C-strain challenge.

### 2.11. E2-mi3 NPs Enhanced Immune Responses in Pigs

As a potential nanovaccine, the protective efficiency assessment of the E2-mi3 NP vaccines was tested in pigs at 28 dpi.

At 28 dpi, peripheral blood lymphocytes (PBLs) were isolated and re-stimulated in vitro with the purified E2-mi3 NP protein to analyze cellular immune responses. As shown in Figure 9A, the stimulation index detected in the groups immunized with E2 or E2-mi3 NPs were numerically higher (*p* < 0.05) than the group with PBS. Interestingly, the values were significantly raised (*p* < 0.05) in the E2-mi3 NP group compared to the E2 group.

PBLs were isolated at 28 dpi, and measured for CD4^+^ and CD8^+^ T-lymphocytes by flow cytometry. As shown in Figure 9B, pigs immunized with E2 and E2-mi3 NPs displayed higher (*p* < 0.05) percentages of CD4^+^ and CD8^+^ T-lymphocytes than the PBS group. Surprisingly, the percentages of CD4^+^ and CD8^+^ T-lymphocytes in the E2-mi3 NP group were dramatically elevated (*p* < 0.05) compared with the E2 group.

IFN-γ secreted by Th1 cells plays critical roles in regulating the cell-mediated immunity, which reflect the antiviral activity of the host. At 28 dpi, PBLs were isolated to perform an ELISpot assay to quantify IFN-γ secreting cells. As shown in Figure 9C, the E2 group and E2-mi3 NP group had obvious numbers of IFN-γ secreting cells in PBLs compared to the PBS group (*p* < 0.05). Importantly, the E2-mi3 NP group stimulated remarkably increased (*p* < 0.05) IFN-γ secreting cells than those of the E2 group.

To analyze the immune responses in the pigs vaccinated with E2 and E2-mi3 NPs, the levels of Th1- and Th2-type cytokines were measured by ELISA. As shown in Figure 9D, the IFN-γ, IL-2, IL-12p70 (Th1-type cytokine), and IL-4 (Th2-type cytokine) levels in the E2 group and E2-mi3 NP group were significantly higher (*p* < 0.05) than those in the PBS group. It is important to note that the contents of IFN-γ, IL-2, IL-12p70, and IL-4 in the E2-mi3 NP group significantly increased (*p* < 0.05) in comparison to the E2 group.

These results indicate that E2-mi3 NP vaccines could trigger stronger cellular and humoral immune responses in pigs than E2 vaccines.

## 3. Discussion

Though conventional inactivated or attenuated live vaccines are effective in the global control or eradication of CSF, they present several drawbacks including challenges in biosafety and the inability to differentiate between natural infection and vaccination, thereby hampering CSF eradication [9,10]. Thus, it is necessary to develop a safer and more effective DIVA vaccine. NP-based technology can be a potent strategy for the generation of rapid and broader effective vaccines that could protect against the constantly emerging pathogens [27,28,29]. Early studies have confirmed that NPs could be used as delivery scaffolds for the targeted display of immunogenic epitopes or proteins to generate chimeric nanovaccines against multiple pathogenic infections [28,29]. In fact, a variety of self-assembled NPs have been proposed such as viral capsid protein and ferritin [30,31]. To understand this concept, we presented a novel strategy to display the CSFV E2 protein on the surface of self-assembled mi3 NPs. Following purification, the E2-mi3 fused protein was found to fold correctly and self-assembled into uniform NPs in vitro, which have an approximate diameter of 30 nm, consistent with early reports [11,20]. Altogether, these data demonstrate that E2-mi3 NPs can be successfully produced using the PichiaPink expression system.

Most biological materials maintain their native features in vitro only in low-temperature environments [32]. For instance, vaccines must be stored and delivered at low temperatures, and the cold chain (spends about 80% of vaccines production cost) has become the critical element that determines vaccine efficacy [33,34]. Hence, improving the thermostability and storage stability of vaccines is an essential step in accelerating massive vaccination. Actually, thermostable vaccines against human enterovirus type 71 have been reported that had a stable morphology (shape and size distribution) after storage for 30 days at 25 °C [35]. Similar to these reports, in our study, self-assembled E2-mi3 NPs exhibited robust thermostability to temperatures ranging from 25 to 70 °C. Even at room temperature (25 °C) for 6 weeks, more than 80% of intact E2-mi3 NPs were still preserved and showed long-term stability. Moreover, we found that the degradation rates of the E2-mi3 NP antigens were significantly slower than the E2 antigens. Based on these results, it can be concluded that the in vitro antigenicity was highly related to homogeneous integrity and revealed that target-mi3 NPs may be an ideal platform for nanovaccine development.

The efficiency with which antigen-presenting cells (APCs) take up antigens is a critical determinant of the elicited immune response. NPs similar to pathogens (10~200 nm) are preferred to uptake by APCs, thereby promoting the initiation of adaptive immune response [36,37]. An early report revealed that NPs show enhanced adsorption and phagocytosis by APCs (such as macrophages) and stimulate their maturation [38]. In the present research, the E2-mi3 NPs were ~30 nm in diameter as shown by the TEM and DLS analyses. Furthermore, the uptake efficiency of the E2-mi3 NP groups was markedly higher than that of the E2 groups. This difference may be attributed to the fact that mi3 NPs enable the multiple adsorption and delivery of E2 antigens into RAW264.7 cells. Our results demonstrate the ability of this NP-based vaccine for elevated antigen uptake.

Considering the promising properties of E2-mi3 NPs as vaccines in vitro, we conducted further assessments of their in vivo protective efficacy through immunization and challenge tests in the rabbit model. Our present study demonstrated that E2-mi3 NPs were closely associated with the triggering of stronger humoral immune responses, particularly in the E2-mi3 NP group, where four out of five rabbits exhibited the ability to neutralize the C-strain viruses at a 1:64 dilution. This suggests that E2-mi3 NP antigens could induce a higher level of humoral immune responses compared to E2 antigens. Furthermore, following CSFV C-strain challenge, no obvious change in CSFV-related clinical symptoms, organ weight, and organ index were detected. Viral antigen distribution (tissues), and viral RNA levels (bloods) were detected in the E2-mi3 NP group, and these findings were consistent with the histopathological observation. Importantly, the E2-mi3 NP vaccines were able to confer complete protection of rabbits from challenge with C-strain (protection rate: 100%). Overall, these results indicate that the E2-mi3 NP vaccines suppressed the production of CSFV in vivo, rendering the rabbits less susceptible to CSFV infection.

Additionally, based on the well-protective efficacy of E2-mi3 NP vaccines in rabbits, we extended our investigation to assess the capability of these NPs in inducing various immune responses in pigs. It is well-known that high titers of neutralizing antibodies play an important role in protection from CSFV challenge [12]. Several studies have demonstrated that a neutralizing antibody titer of 32 conferred protection against CSFV infection [8,39,40]. In our present study, the levels of specific antibodies and neutralizing antibodies were remarkably raised by E2-mi3 NPs as compared to E2; peculiarly, the E2-mi3 NP group pig serum (4/5) could completely neutralize the viruses at the 1:64 dilution. Based on the above results, we speculated that E2-mi3 NPs could offer good protection of pigs against CSFV challenge. In addition, certain non-antibody-mediated immune mechanisms such as T-cell activation and the cytokines they release may also play a role for conferring early protection against CSFV [41,42]. The level of T-lymphocyte proliferation reflects the overall immune state of the body [42,43]. In this study, the stimulation index was significantly raised in the E2-mi3 NP group compared to the E2 group, suggesting that the E2-mi3 NPs were able to trigger stronger cellular immune responses. IFN-γ secreted by Th1 cells plays momentous roles in regulating the cell-mediated immunity, which reflects the antiviral activity of the host [41,42]. In our present study, E2-mi3 NPs stimulated remarkably increased numbers of IFN-γ secreting cells than those of the E2, indicating that the E2-mi3 NPs significantly improved the protective immune responses in the elimination of intracellular pathogens. CD4^+^ and CD8^+^ T cells are two major lymphocyte subsets in adaptive immune responses. After being activated, CD4^+^ T cells differentiate into different subsets called Th1, Th2, and Th17 cells, which activate B cells to elicit immune responses [43,44,45]. CD8^+^ T cells are mainly involved in the cellular immune response, which represent an important defense mechanism in the elimination of cells infected by CSFV [43,45]. It is important to note that, in the present research, the percentages of CD4^+^ and CD8^+^ T-lymphocytes in the E2-mi3 NP group were dramatically elevated compared with the E2 group, which suggest that the E2-mi3 NPs could induce higher immune responses in pigs than E2. Cytokines associated with the Th1 immune responses (IFN-γ, IL-2, and IL-12p70) are reported to contribute to the activation of CD8^+^ T cells, which trigger cellular immune responses [46,47]. IL-4 is a Th2-type cytokine that can promote the proliferation and activation of B cells, and is a key regulator of humoral and adaptive immune responses [47]. In the present study, the levels of IFN-γ, IL-2, IL-12p70, and IL-4 in the E2-mi3 NP group significantly increased in comparison to the E2 group, suggesting that E2-mi3 NPs were easier to activate cellular and humoral immune responses. To sum up, these data provide supporting evidence that E2-mi3 NP antigens can induce a higher level of immune responses compared to E2 antigens.

Interestingly, based on the above study results, we found that antibodies against mi3 NPs did not interfere with the antibodies targeting E2, thus reducing the potential side effects caused by anti-carrier immunity [48]. It should be noted that mi3, derived from an artificial proteinaceous self-assembled molecule, could not cause any side effects or safety concerns, which has been described in previous reports [20]. Moreover, in order to further verify this observation, the locations of the conformational epitopes in the CSFV E2 protein and E2-mi3 protein were analyzed using the Pymol 2.5 software. As expected, the main parts of the E2 protein and E2-mi3 protein (except the skeleton protein of mi3) overlapped almost completely, and the docking location of the E2 protein and mi3 skeleton protein was not on the antigenic epitopes of the E2 protein. These results clearly demonstrate that the E2 protein has a certain affinity with the mi3 skeleton protein, and that the mi3 skeleton protein has no effect on the antigenic epitopes of the E2 protein. This finding provides supporting evidence that self-assembled mi3 NPs allow for the targeted display of antigens and that NP-based delivery technology can be a potent strategy for nanovaccine development.

In conclusion, our current results show the potential of using CSF NPs produced in yeast as a vaccine candidate. These greatly stable, highly immunogenic, and structure-based, self-assembled NPs could accelerate the broad application of self-assembled NPs for vaccine development.

## 4. Materials and Methods

### 4.1. Cloning

Gene sequences coding for E2, mi3, and E2-mi3 were synthesized and cloned into the yeast (PichiaPink) expression vector pPink-HC to construct the expression plasmid pPink-E2, pPink-mi3, and pPink-E2-mi3, respectively. Flexible linker was incorporated between the E2 protein and mi3 to facilitate proper folding. All plasmids were constructed using standard methods and verified by DNA sequencing. Recombinant yeasts were obtained as described by the manufacturer’s instructions.

### 4.2. Expression and Purification of Recombinant Protein

The recombinant proteins (E2, mi3, or E2-mi3) were expressed and purified as described previously [49]. Then, the supernatants of the recombinant proteins were analyzed by sodium dodecyl sulfate polyacrylamide gel electrophoresis (SDS-PAGE) and Western blot. The assembled NP size was measured by dynamic light scattering (DLS) with a Zetasizer-Nano (Malvern Zetasizer Nano ZS90; Worcestershire, UK). The morphology of the NPs were observed by transmission electron microscopy (TEM) (HT7700; Hitachi, Tokyo, Japan) after dyeing with phosphotungstic acid.

### 4.3. Cells and Viruses

RAW264.7 (mouse leukemia cells of monocyte macrophage) cells were cultured in Dulbecco’s modified Eagle’s medium (Gibco, Carlsbad, CA, USA) supplemented with 10% fetal bovine serum. The CSFV C-strain was stored in our laboratory.

### 4.4. The 3D Structure Analysis of CSFV E2-mi3

The 3D structure of E2 and E2-mi3 of the CSFV was modeled using the Alphafold2-2.3.1 server (https://github.com/deepmind/alphafold) (accessed on 15 May 2023). The locations of the identified conformational epitopes on the predicted model of the E2 and E2-mi3 protein of CSFV were revealed by the Pymol 2.5 software (https://www.pymol.org) (accessed on 15 May 2023). 

### 4.5. Thermostability and Long-Term Storage Stability Evaluation of NPs

To evaluate thermostability under different conditions, stocks of the E2-mi3 NP protein and E2 protein were incubated at 4, 25, 37, 50, 60, or 70 °C for 1 h and then cooled to 4 °C for 10 min. Subsequently, the protein samples were centrifuged at 16,000× *g* for 30 min at 4 °C. The supernatant was analyzed by SDS-PAGE and determined by densitometry. Hydration particle size change of the E2-mi3 NP protein was detected by DLS. The sample held at 4 °C was defined as 100% soluble, as described in a previous study [21].

As for the long-term storage stability analysis, stocks of the E2-mi3 NP protein were aseptically filtered and incubated at room temperature (25 °C). The samples were collected every two weeks and centrifuged to remove aggregates, and subjected to TEM for E2-mi3 NP integrity analysis. Then, the numbers of E2-mi3 NPs were used to count at 25,000× magnification of 5 fields/image using ImageJ2x (1.4.3.67) software, as previously described [11]. The sample held at 0 week was defined as 100%. In addition, the effective antigens were quantitatively measured by Dot blotting for long-term storage stability analysis [44].

### 4.6. Cellular Uptake Assay

RAW264.7 cells were stimulated by the E2-mi3 NPs protein (10 μg) and E2 protein (10 μg), respectively, and then incubated under an atmosphere of 5% CO_2_ at 37 °C for 1, 2, 4, 6, and 8 h. At the indicated time points, the samples were collected and detected by Western blot. Statistical analysis of protein expression was performed using ImageJ2x (1.4.3.67) software.

### 4.7. Animal Experiment

Vaccines were prepared by antigens (E2-mi3 NPs and E2) with ISA-201 adjuvant (SEPPIC) (1:1, *w/w*) according to the manufacturer’s manual.

For rabbit immunization and challenge, 8-week-old CSF antibody negative New Zealand White Rabbits were randomly divided into three groups with five animals per group. The E2-mi3 NP group and E2 group were intramuscularly inoculated with 10 μg E2-mi3 NPs or 10 μg E2, respectively. The same amount of PBS was given to the rabbits in the PBS group by the same method. All rabbits were challenged intravenously with C-strain (0.1 mL/per rabbit; 10^6^ TCID_50_/mL) at 28 dpi.

For swine immunization, fifteen 15-week-old CSF antibody negative pigs were randomly assigned to three groups: the E2-mi3 NP group and E2 group were injected intramuscularly with 40 μg E2-mi3 NPs or 40 μg E2, respectively. Pigs in the PBS group were injected with PBS. The rabbits and pigs were observed daily for clinical symptoms, and rectal temperatures were measured. Fever was defined as a rectal temperature > 39.5 °C.

### 4.8. Determination of Specific Antibodies

Serum samples were collected from rabbits and pigs at 7, 14, 21, and 28 dpi and then the following tests were performed: ELISA plates were precoated with 100 μL/well of mi3 protein at the concentration of 500 ng and incubated at 4 °C overnight. Subsequently, the levels of mi3 protein specific antibodies were detected through indirect-ELISA. The levels of CSFV E2 specific antibodies were detected through the classical swine fever virus antibody test kit (IDEXX Switzerland AG, Bern, Switzerland) according to the manufacturer’s instructions. CSFV antibody positive: sample blocking ratio% ≥ 40%. The blocking ratio of the serum sample was calculated using the following formula: Blocking ratio% = (NC OD450 nm − Sample OD450 nm)/ (NC OD450 nm) × 100%. 

### 4.9. Rabbit Neutralization Test

The rabbit body neutralization test was carried out with a fixed dose of rabbit virus and diluted serum. The serums collected on the 28 dpi were diluted with physiological saline at a ratio of 1:4, 1:16, and 1:64 and mixed with the C-strain of CSFV for 2 h at 4 °C. After completing the neutralization of the serum and virus, each rabbit in the experimental group was injected with 1 mL of the serum–virus mixture (0.1 mL C-strain/per rabbit; 10^6^ TCID_50_/mL) through the ear vein, while each rabbit in the PBS group was injected with 1 mL of physiological saline–virus mixture (0.1 mL C-strain/per rabbit; 10^6^ TCID_50_/mL) by the same method. Rectal temperatures were recorded every 6 h post-challenge, and continuously observed for 72 h. Result judgment: After the rabbit was challenged, the rabbit showed a fixed thermal response (rectal temperature greater than 39.5 °C and lasting for at least 12 h), and the rabbit was judged as fever (+), otherwise, it was judged no fever (—).

### 4.10. Histopathological Assessment 

The tissues (hearts, livers, spleens, lungs, and kidneys) were collected from rabbit at 4 days post-challenge. After necropsy, whole tissues (hearts, livers, spleens, lungs, and kidneys) were removed aseptically from individual animals, weighed, and photographed. The organ index was calculated using the following formula: Organ index = (Organ weight)/(Body weight) × 100%.

The tissues (hearts, livers, spleens, lungs, and kidneys) were collected and then immediately fixed in 4% paraformaldehyde overnight. Subsequently, the tissues were dehydrated through graded alcohol, paraffin embedded, sectioned at 5 μm, and processed for hematoxylin and eosin staining. Histopathological changes were observed and photographed with a digital camera under 400× magnifications (Nikon DS-Ri1, CHANSN INSTRUMENT (SHANGHAI) CO., LTD, Shanghai, China). The same position of the hearts, livers, spleens, lungs, and kidneys in five rabbits was observed through a microscope, and the histological lesions of the heart, liver, spleen, lung, and kidney were evaluated through the incidence of congestion and hemorrhage as well as the severity scoring of inflammation infiltrate. The level of severity was judged from − to ++++, which represented none to severe. The histological lesions were scored as described in a previous study [50,51].

### 4.11. Immunohistochemical (IHC) Staining and Analysis

The paraffin sections were treated with 3.0% hydrogen peroxide followed by boiling sodium citrate solution and incubated overnight with the CSFV E2 monoclonal antibody at 4 °C. Then, the sections were incubated with an HRP-conjugated anti-pig IgG antibody for 1 h at 37 °C. Finally, the results were visualized by DAB. The proportions of the IHC-positive cell area were calculated at 200× magnification of 5 fields/image using the HALO digital pathology system (Halo 101-WL-HALO-1, Indica labs, Albuquerque, NM, USA) as previously described [52]. Result judgment: Brown granules were positive cells. 

### 4.12. Real Time Quantitative PCR (RT-qPCR) Assay for Detection of Viral RNA Loads

The anticoagulated blood samples were analyzed by RT-qPCR to determine viremia. Viral RNA was extracted from the rabbit anticoagulated blood with TRIzol reagent (Invitrogen, Waltham, MA, USA) according to the manufacturer’s instructions. Then, the viral RNA loads were assessed with a PrimeScript One-Step RT-PCR Kit (TaKaRa, Dalian, China) and primers specific for the CSFV IRES (5′-TAACACCACCGTAAAAGTAC-3′ and 5′-TTCCCCCTAATGCCACT-3′). Samples were considered negative when the CSFV RNA loads/500 μL were less than 10^3^. The viral RNA loads were analyzed by RT-qPCR, as previously described [53].

### 4.13. Lymphocyte Proliferation Assay

At week 4 of immunization, anticoagulated blood samples were collected from pigs and peripheral blood lymphocytes (PBLs) were aseptically isolated in accordance with the manufacturer’s instructions. Isolated lymphocytes were resuspended in RPMI-1640 medium supplemented with 10% FBS (*v*/*v*) and 1% penicillin-streptomycin solution (*v/v*). The cells were counted and diluted to 10^6^ cells/well and transferred to 96-well plates for the lymphocyte proliferation assay. Then, 1 μL concanavalin A (Sigma-Aldrich, St. Louis, MO, USA), and CSFV E2 protein were added to the wells as a positive control, and a blank control group containing only culture medium but not cells was established. After 72 h of incubation, 10 μL MTS reagent was added into each well, and the plates were incubated for 4 h at 37 °C in 5% CO_2_. The OD values were measured at 490 nm using a spectrophotometer, and the stimulation index (SI) was calculated as follows: SI = (stimulation group − blank control group)/ (negative control group − blank control group).

### 4.14. Analysis of CD4^+^ and CD8^+^ T-Lymphocytes

At 28 dpi, PBLs were collected from each pig. One million PBLs were separated and transferred into a 1.5 mL centrifuge tube. One milliliter of a fluorescence solution was then added and underwent centrifugation. The supernatant was removed, and the cell pellet was resuspended in 500 μL of cell fluorescence solution for staining with anti-pig CD3-Percp-CY505, anti-pig CD4-PE-CY7, and anti-pig CD8a-Biotin (Becton, Dickinson and Company, Franklin Lakes, NJ, USA) fluorescent antibodies at 4 °C in the dark for 0.5 h. After washing twice with the fluorescence solution and centrifugation, the supernatant was discarded. The cell pellet was resuspended in 500 μL of fluorescence preservation solution. Flow cytometry (Becton, Dickinson and Company, USA) was then used to count CD3^+^ CD4^+^ and CD3^+^ CD8^+^ T-lymphocytes in 30,000 cells, and percentages of CD3^+^ CD4^+^ and CD3^+^ CD8^+^ T-lymphocytes were determined. 

### 4.15. Interferon (IFN)-γ Detection by ELISpot

The numbers of IFN-γ-secreting cells in porcine PBLs were quantified by an ELISpot kit (MABTECH) at 28 dpi. In the ELISpot assay, each spot represents a cell that secretes IFN-γ. The cells that formed spots were called spot forming cells (SFCs). Data are presented as mean numbers of antigen-specific IFN-γ-secreting cells per 10^6^ PBLs from duplicate wells of each sample. The IFN-γ ELISpot assay was carried out as previously described [45,54].

### 4.16. Cytokines Measurement by Enzyme-Linked Immunosorbent Assay (ELISA)

Serum IFN-γ, interleukin (IL)-2, IL-4, and IL-12p70 at 28 dpi were measured with porcine ELISA kits (NeoBioscience Technology Company Limited, Shanghai, China) according to the manufacturer’s instructions.

### 4.17. Statistical Analysis

All of the data were analyzed by SPSS 22.0 software. All of the results were expressed as the mean ± standard deviation (SD). The significance of difference was analyzed by the independent samples *t* test between two groups, or by variance analyses (LSD or Dunnett’s T3) among the three groups. Statistical significance was considered at *p* < 0.05.

## Figures and Tables

**Figure 1 ijms-25-00596-f001:**
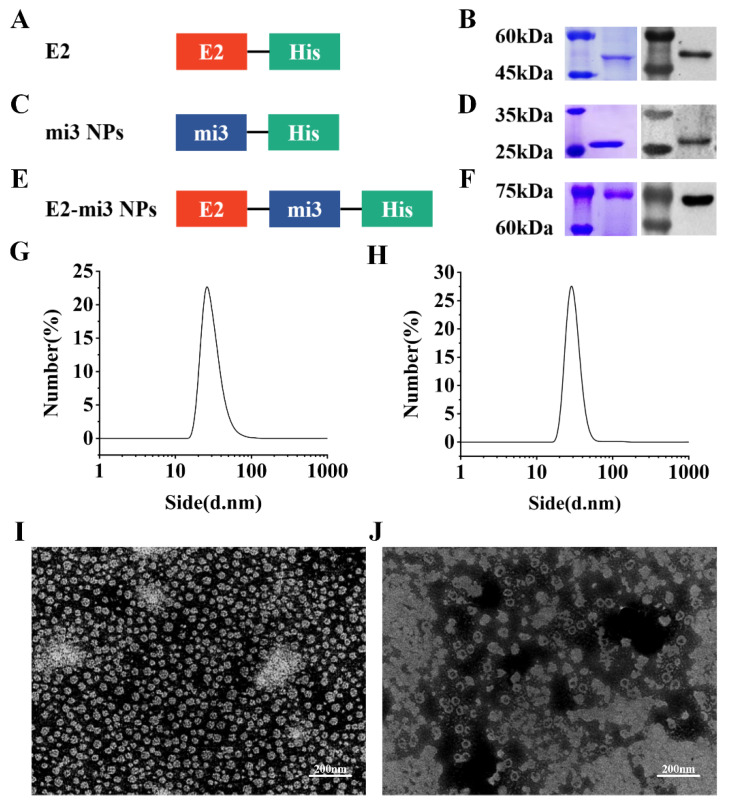
Expression and self-assembly analysis of CSFV E2-mi3 NPs. Diagram of plasmid E2 (**A**), mi3 NPs (**C**), and E2-mi3 NPs (**E**). SDS-PAGE (**left**) and Western blot (**right**) analysis of the purified E2 protein (**B**), mi3 NPs protein (**D**), and E2-mi3 NPs protein (**F**). The size distribution of the mi3 NPs (**G**) and E2-mi3 NPs (**H**) was analyzed by DLS. TEM analysis of self-assembly capacity of mi3 NPs (**I**) and E2-mi3 NPs (**J**), scale bar = 200 nm.

**Figure 2 ijms-25-00596-f002:**
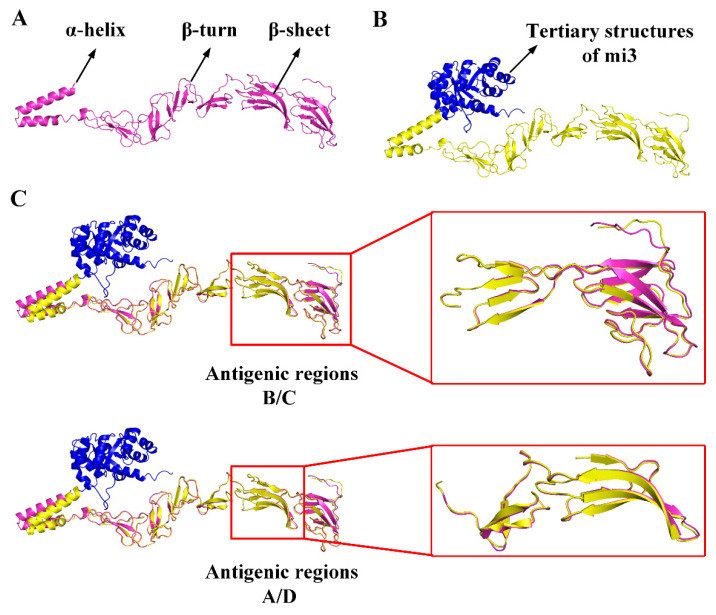
Cartoon of the tertiary structure modeling of the CSFV E2 protein and E2-mi3 protein. (**A**) 3D model of the CSFV E2 protein. (**B**) 3D model of the CSFV E2-mi3 protein (mi3 skeleton protein tertiary structures are shown in blue). (**C**) The 3D model of the CSFV E2 protein and E2-mi3 protein were analyzed by Pymol.

**Figure 3 ijms-25-00596-f003:**
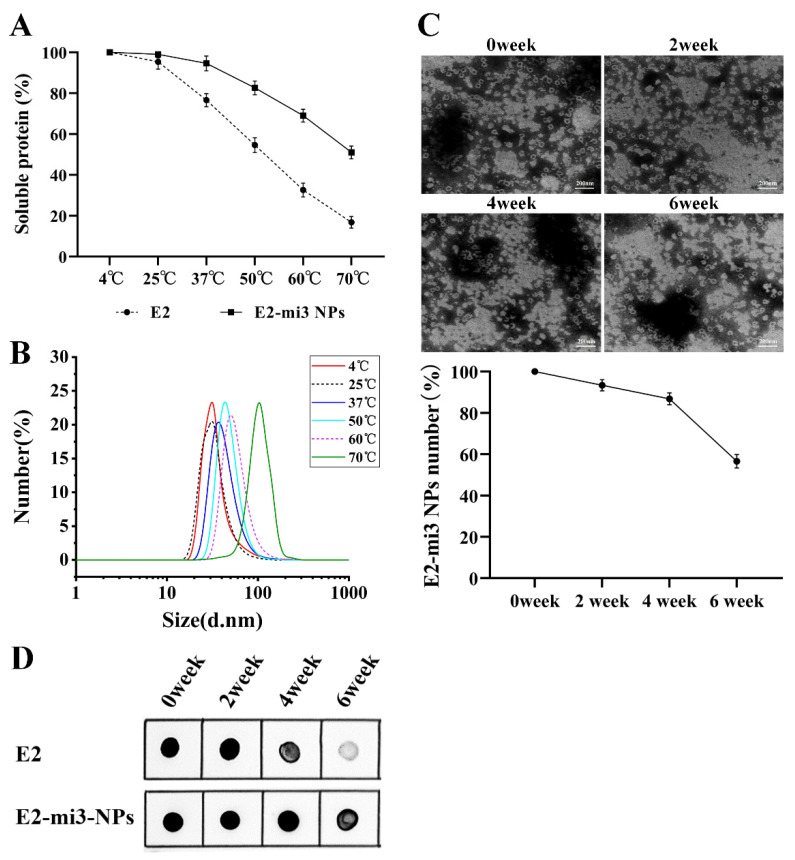
Thermostability and long-term storage stability evaluation of the CSFV E2-mi3 NPs. (**A**) E2-mi3 NP protein remained soluble after high temperatures. (**B**) Size distribution of the E2-mi3 NP protein at different temperatures. (**C**) E2-mi3 NP integrity following long-term storage. ImageJ2x (1.4.3.67) software was used to count the number of E2-mi3 NPs. (**D**) Effective antigens analysis of the E2 protein and E2-mi3 NPs protein by Dot blotting.

**Figure 4 ijms-25-00596-f004:**
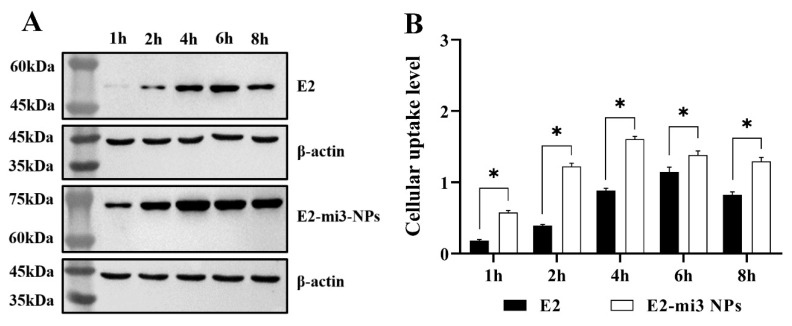
Cellular uptake efficiency evaluation of CSFV E2-mi3 NPs. (**A**) The amount of E2 protein and E2-mi3 NP protein in RAW264.7 cells was measured by Western blot. (**B**) Gray value analysis of RAW264.7 cells. Note: * *p* < 0.05.

**Figure 5 ijms-25-00596-f005:**
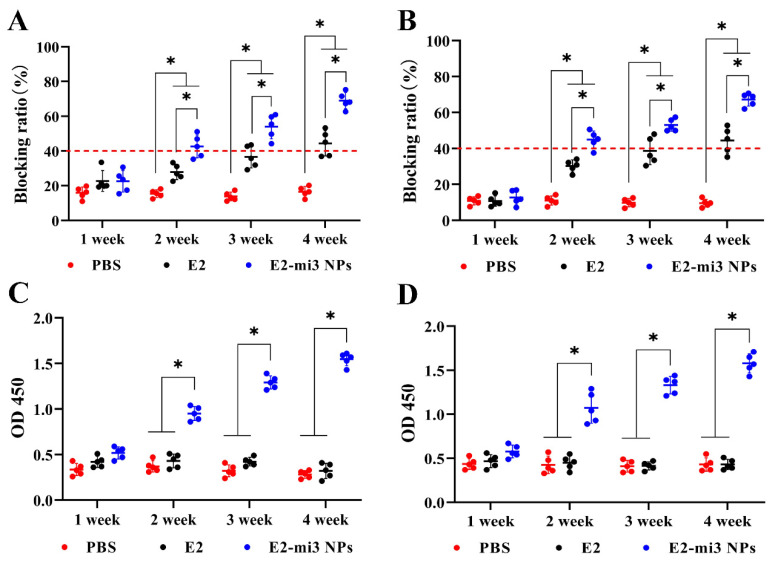
Specific antibody responses induced by E2-mi3 NPs in rabbits and pigs. Rabbit serum samples for the detection of CSFV-specific antibodies (**A**) through the CSFV antibody test kit and mi3 protein specific antibodies (**C**) with indirect-ELISA. Pig serum samples for the detection of CSFV-specific antibodies (**B**) via the CSFV antibody test kit and mi3 protein specific antibodies (**D**) through indirect-ELISA. Note: * *p* < 0.05.

**Figure 6 ijms-25-00596-f006:**
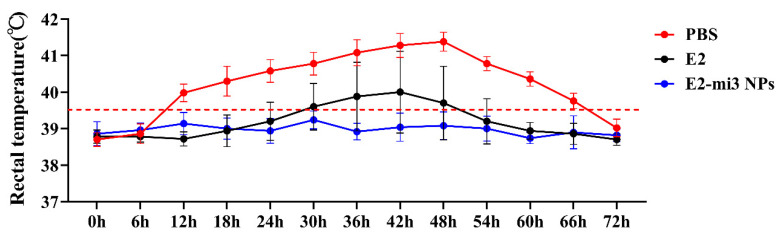
Protective efficacy of E2-mi3 NPs in rabbits. Note: Rectal temperatures of the rabbits following CSFV C-strain challenge.

**Figure 7 ijms-25-00596-f007:**
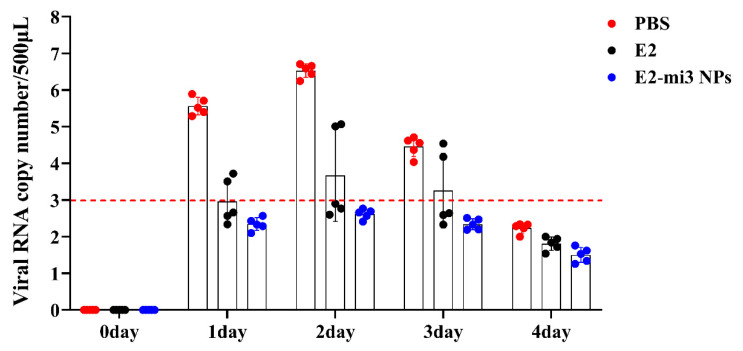
Detection of viral RNA in the blood samples from the immunized rabbits after being challenged with the C-strain by RT-qPCR (viral RNA values were obtained by Log^10^ processing). Note: Samples were considered negative when CSFV RNA loads/500 μL were less than 10^3^.

**Figure 8 ijms-25-00596-f008:**
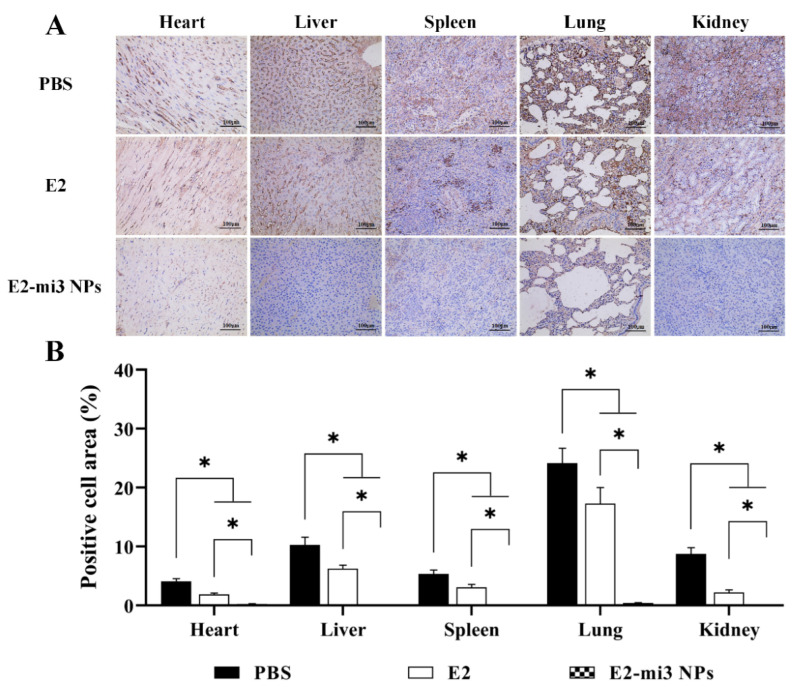
Immunohistochemical analysis of viral replication in rabbits after challenge. (**A**) Representative immunohistochemical examination of immunized rabbits challenged with the CSFV C-strain (scale bar = 100 μm). (**B**) The proportions of positive cell area in these tissues (heart, liver, spleen, lung, and kidney). Note: * *p* < 0.05.

**Figure 9 ijms-25-00596-f009:**
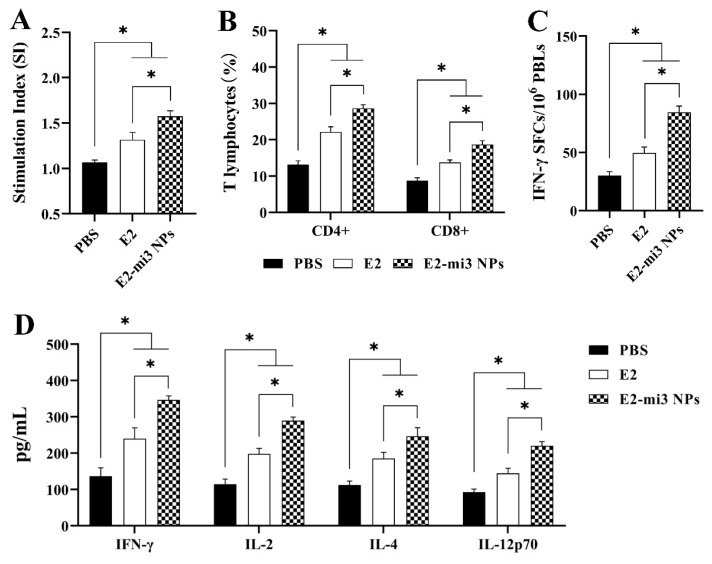
Humoral and cellular immune responses in pigs induced by E2-mi3 NPs. (**A**) Lymphocyte proliferation assay was performed to assess the cellular immune responses. (**B**) The percentages of CD4^+^ and CD8^+^ T-lymphocytes in peripheral blood lymphocytes were analyzed by flow cytometry. (**C**) ELISpot assay of IFN-γ secreted by porcine peripheral blood lymphocytes. (**D**) Cytokine levels (IFN-γ, IL-2, IL-4, and IL-12p70) of sera from pigs immunized were measured by ELISA. Note: * *p* < 0.05.

**Table 1 ijms-25-00596-t001:** Rabbit neutralization test.

Groups	Number	Rabbit Serum			Pig Serum		
		1:4	1:16	1:64	1:4	1:16	1:64
E2-mi3 NPs	1	—	—	—	—	—	—
	2	—	—	—	—	—	—
	3	—	—	—	—	—	—
	4	—	—	—	—	—	—
	5	—	—	+	—	—	+
E2	1	—	—	+	—	—	+
	2	—	—	+	—	—	+
	3	—	—	+	—	—	+
	4	—	+	+	—	+	+
	5	—	+	+	—	+	+
PBS	1	+			+		
	2	+			+		
	3	+			+		
	4	+			+		
	5	+			+		

Note: “+”: Fever; “—”: No fever.

**Table 2 ijms-25-00596-t002:** Protection of rabbits challenged with C-strain.

Groups	Number	Counts of Fever Reactions	Rate of Protection (%)
PBS	5	5/5	0
E2	5	2/5	60
E2-mi3 NPs	5	0/5	100

## Data Availability

The data presented in this study are available on request from the corresponding author. The data are not publicly available due to intellectual property considerations.

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
