# Peer review of "Self-Assembling E2-Based Nanoparticles Improve Vaccine Thermostability and Protective Immunity against CSFV"

_ijms, 2024, doi:10.3390/ijms25010596_

Round 1

Reviewer 1 Report

Comments and Suggestions for Authors

The authors have presented a self-assemblying nanoparticle vaccine based on the E2 glycoprotein. The results were substantial and showed good protective efficacy in all relevant manners. I have a few questions that I would like to ask:

1. The use of the dotblot to determine thermostability is relatively rough. Is there a better way for the author to do so? Possibly a western blot with an SDS page to show the changes/differences in the protein size? A smearing or reduction in smearing could improve the claim?
2. WIth regards to its thermostability, were the authors able to use a "stored" vaccine to compare with a freshly made vaccine in order to investigate the loss of activity in an immunization experiment? That would be able to provide functional changes to the vaccine and effects of the change? 
3. The data pertaining the neutralization done in rabbits were very conclusive but the challenge was done intravenously but CSF occurs through the mucosal/oral route. Did the authors try to investigate other antibodies especially those in the oral/mucosal route? Will it be possible to have challenge studies based on oral/mucosal delivery route?
4. Would it be possible for the author to discuss the possibility of converting the vaccine into a oral/mucosal vaccine as the nanoparticles are already stable in high temperature to be converted either in a freeze drying manner or some other manner of powdering the vaccine which can then be delivered via those routes.

Thank you

Comments on the Quality of English Language

Some of the sentences are very lengthy and would require better punctuation. It would be better to reduce the use of exaggerating terms such as "remarkably" and etc.   

Reviewer 2 Report

Comments and Suggestions for Authors

This manuscript from Song et al details the production and characterization of a novel nanoparticle-based vaccine against classical swine fever virus (CSFV).  While current vaccines protect against severe and widespread disease in pigs caused by CSFV, eradication of CSFV is limited in part due to an inability to serologically distinguish between infected and vaccinated animals.  To address this challenge, the authors generated a novel CSFV vaccine based on the mi3 self-assembling nanovaccine platform, which was fused to the CHSV E2 protein, which is the antigenic component of most existing CHSV vaccines.  mi3-E2 was successfully expressed in PichiaPink yeast cells, and was found to be thermodynamically stable and to increase cellular uptake of E2 in-vitro.  When tested in rabbits and mice, mi3-E2 produced significant higher levels of anti-E2 antibodies (which were also found to be neutralizing) than E2 alone, despite the co-production of anti-mi3 antibodies.  mi3-E2 was also found to protect against fever, tissue pathology and significant viremia in rabbits, as well as having significantly higher T cell and cytokine activity.

While the design and characterization of mi3-E2 is solid, I have some comments to be addressed to strengthen the study design, particularly for the in-vivo experiments.  Some areas listed below also need more information on study design or editing for clarification.

1) In the introduction, the authors should indicate whether previous studies have found challenges with making VLP-E2 vaccines (as was brought up in paragraph 3).  The authors should also indicate whether VLP vaccines have been successfully used as a marker for DIVA and how long this signal can be detectable after vaccination.

2) In figure 1A, the authors should include sizes for the constructs (including each of the protein components) to correlate with the protein blots in 1B.

3) If available, the authors should provide supplemental information to the structural analysis performed on alphafold, such as optimization for any environmental conditions.

4) In section 2.2, the authors should provide citations for the antigenic epitopes found on E2.

5) In figure 2C, the authors should highlight which residues on E2 and mi3 are predicted to interact.

6) In section 2.3, the authors should highlight how the demonstrated thermostability of mi3-E2 compares to other nanoparticle vaccines.

7) In figure 3C, the authors should clarify how they determined the total number of E2-mi3 aggregates to obtain the percentage of aggregates that were intact (ie- how were individual non in-tact E2-mi3 aggregates counted).

8) Figure 3D would benefit from running a western blot in addition to dot blot to determine whether E2 signaling can be picked up from partially degraded E2-mi3 aggregates.

9) For clarification, the authors should also include the dosage and vaccination schedule in the text for section 2.5 (which is currently just included in the methods).

10) The authors should include more details on their statistical analysis besides indication for p<.05 (using the standard metrics for *, ** and *** designations of significance), as many of these comparisons look to have higher degrees of significance.

11) The first paragraph in section 2.6 as well as its corresponding inclusion in the discussion section (page 14 paragraph 3) is confusingly worded and should be edited for clarity.  The paragraph would also benefit from including the method used for viral neutralization (as opposed to a cell-based assay).

12) If possible, the authors should also test the ability of mi3-E2 elicited antibodies to reduce or eliminate viremia to supplement Table 1, either in-vivo or in-vitro.

13) While the authors include data for weight loss in figure S1, its significance is hard to determine due to weight loss only being included for pre and post challenge, when weight gain would resume in infected animals shortly after their fevers reside.  If possible, the authors should instead weigh animals daily after challenge infection to supplement Figure 6.

14) Timepoints of sample collection need to be included for figures 8, S1 and S2.

15) The authors should also quantify tissue sections in figure 8 by total intensity of HRP signal/ area to account for the low level of signal seen in the E2-mi3 animals (particularly in the lung).

16) Each tissue in figure S1C and D should be shown in separate graphs to account for differences in scaling between each organ.

17) The authors should clarify how they calculated organ index in figure S1D.

18) In figure 9B, the authors should report % of T cells as a % of total lymphocytes rather than total cells to account for changing lymphocyte numbers post vaccination, or should report absolute counts when normalized to a counting standard.  The authors should also report the ratio of CD4:CD8 T cells to suggest a Th1/Th2 bias (if Th1/2 cells were not measured directly).

Comments on the Quality of English Language

Minor corrections needed for final edit- sections of most need were noted in the main review.

Round 2

Reviewer 2 Report

Comments and Suggestions for Authors

Comments have been sufficently addressed for publication

Author Response

We have addressed the issue in the revised manuscript.